# Surgical and Medical Aspects of the Initial Treatment of Biliary Atresia: Position Paper

**DOI:** 10.3390/jcm11216601

**Published:** 2022-11-07

**Authors:** Mark Davenport, Omid Madadi-Sanjani, Christophe Chardot, Henkjan J. Verkade, Saul J. Karpen, Claus Petersen

**Affiliations:** 1Department of Paediatric Surgery, Kings College Hospital, London SE5 9RS, UK; 2Klinik für Kinderchirurgie, Medizinische Hochschule Hannover, Carl-Neuberg-Str. 1, 30625 Hannover, Germany; 3Chirurgie Pédiatrique—Transplantation, Hôpital Necker—Enfants Maladies, Université Paris Descartes, 149 Rue de Sèvres, 75015 Paris, France; 4Center for Liver, Digestive and Metabolic Diseases, Universitair Medisch Centrum, 9713 AV Groningen, The Netherlands; 5Center for Advanced Pediatrics, 1400 Tullie Circle SE 2nd Floor, Atlanta, GA 30329, USA

**Keywords:** biliary atresia, Kasai operation, adjuvant therapy, corticosteroids, cytomegalovirus, ursodeoxycholic acid

## Abstract

Biliary atresia, a fibro-obliterative disease of the newborn, is usually initially treated by Kasai portoenterostomy, although there are many variations in technique and different options for post-operative adjuvant medical therapy. A questionnaire on such topics (e.g., open vs. laparoscopic; the need for liver mobilisation; use of post-operative steroids; use of post-operative anti-viral therapy, etc.) was circulated to delegates (*n* = 43) of an international webinar (Biliary Atresia and Related Diseases—BARD) held in June 2021. Respondents were mostly European, but included some from North America, and represented 18 different countries overall. The results of this survey are presented here, together with a commentary and review from an expert panel convened for the meeting on current trends in practice.

## 1. Introduction

Biliary atresia (BA) is an obliterative condition of the biliary tract that typically presents with jaundice and pale stools during neonatal life [1,2]. All affected infants have conjugated hyperbilirubinemia, often evident in the first days of life [3], along with elevated liver enzymes, but may present little in the way of conclusive diagnostic signs, at least in the first few weeks of life. Although diagnosis can be strongly suggested by a range of secondary investigations including abdominal ultrasound, radionuclide imaging, and liver biopsy, it is usually confirmed only at surgical exploration and intraoperative cholangiography.

This investigative strategy has changed little over the past few years, although there is emerging interest in newer and potentially more specific biochemical markers such as MMP-7 [4]. To date, the early studies have been Asian in origin and retrospective [5], with conflicting cut-off values. Other possible diagnostic methods, again still very much in the investigative phase, include the use of AI algorithms in the interpretation of ultrasound data [6].

It is possible to screen for cases of BA, and this has been national strategy in Taiwan and Japan, based on relatively low-tech stool-colour cards. More recently, particularly in Texas, whole blood sampling with measurement of conjugated bilirubin has been trialled [7].

The definitive management of this condition is entirely surgical, usually with an initial attempt at restoration of bile flow and preservation of the native liver (Kasai portoenterostomy—KPE) in the first few months of life. If this fails or is felt to be futile, liver transplant may be carried out. In 1959, Morio Kasai, a Japanese surgeon working in Sendai, described his original experience with what has become known as KPE [8]. Over the subsequent two decades, this operation and its principle—radical excision of the extrahepatic ducts with a biliary reconstruction in the porta hepatis—gradually became accepted for infants with BA, although many modifications (e.g., Roux loop stomas and intussusception valves) were attempted along the way [9,10]. It is also possible to perform a KPE laparoscopically, although this is controversial and has not been widely adopted outside certain high-volume Asian centres [11,12].

The first ever liver transplant in a child born with BA was reported in 1963 by Thomas Starzl and a team from Denver, CO, USA [13]. Although actually unsuccessful, it prompted the first wave of liver transplant centres to be set up during the 1960s, which demonstrated the validity of the surgical technique. Nonetheless, the lack of effective immunosuppression precluded longer-term success and a moratorium was declared. Liver transplantation was re-invigorated in the 1980s with the discoveries of cyclosporin and subsequently tacrolimus, leading to its widespread adoption throughout the world.

The aim of this paper is to review the current diagnostic strategies of BA, and to provide a review of current operative techniques and the role of post-operative adjuvant therapy for KPE, aligned to a survey of current practice. The future direction of clinical research was also explored.

## 2. Methods

An online survey was conducted among members of the European reference network RARE-LIVER, and members of the faculty of the Biliary Atresia and Related Diseases (BARD) network (Appendix A). Questions were drafted in multiple choice format by members of the working group, all of whom were experienced paediatric hepatologists and hepatobiliary surgeons. The survey was performed via an online tool (SurveyMonkey, Survey Monkey Inc., now Momentive Inc.) and was completed anonymously. Respondents could not be traced back to any participating centre. The results of the questionnaire were subsequently discussed by an online panel during the 2021 online Biliary Atresia and Related Diseases (BARD) conference.

## 3. Results

Completed forms were analysed from 43 respondents (with 22 self-declared as surgeons) from centres distributed across 18 countries.

### 3.1. Diagnostic Strategy

In patients with BA, the age at which KPE is performed is negatively related to the success rate of the surgery, in terms both of clearance of jaundice and survival with native liver [14,15]. However, neonatal cholestasis can be caused by a multitude of other conditions and diseases other than BA, for which surgery is not indicated. It has been estimated that BA accounts for 25–40% of neonatal cholestasis [16]. Accordingly, the initial diagnostic strategy in infants with neonatal cholestasis aims to rapidly identify whether a condition other than biliary atresia is the cause of the cholestasis. The workup of neonatal cholestasis has been summarised in the joint guidelines of the North American Society for Pediatric Gastroenterology, Hepatology, and Nutrition (NASPGHAN) and the European Society for Paediatric Gastroenterology, Hepatology, and Nutrition (ESPGHAN); the reader is directed there for details [16]. The causes of neonatal cholestasis can be categorized in anatomical extrahepatic obstructions of bile flow (such as choledochal malformations, cholelithiasis), genetic diseases, either multisystemic (Alagille syndrome, cystic fibrosis, galactosemia, mitochondrial diseases, and others) or exclusively hepatic (alpha-1-antitrypsin deficiency, bile acid synthesis defects, canalicular membrane transport protein defects, and others), endocrinologic disorders (hypocortisolism, hypothyroidism), or as secondary to other diseases (e.g., sepsis, congenital viral infection). Over recent years, the possibilities for genetic analyses have increased [17,18]. The time needed for genetic analysis has so far precluded it from being a prerequisite before intraoperative cholangiography and, in case of BA, KPE surgery. Since the turn-around time for genetic analyses has decreased over the past decade, this may change in future. Early candidates would then be expected to be infants referred in the very first weeks of life, who may not yet have acholic stools.

Generally, the diagnostic strategy for neonatal cholestasis aims at demonstrating or excluding the most frequent non-BA causes. The workup as indicated in the guidelines includes (re)checking of newborn blood-screening results (galactosemia, hypothyroidism, cystic fibrosis), and blood analyses including measurement of white blood counts, differential alpha-1-antitrypsin level and phenotype, thyroid hormone and TSH, bile acid concentration, cortisol, glucose, lactate, and metabolic parameters. Urine analysis is aimed at reducing substances (galactitol), and bacterial or viral infections (including CMV PCR). Imaging of the liver is performed by ultrasound for anatomical abnormalities (choledochal malformations, cholelithiasis) and by X-ray for evidence of multisystem disease (such as in Alagille syndrome, or butterfly vertebrae). On indication such as a cardiac murmur, a cardiac ultrasound doppler is performed for detection of congenital malformations, such as pulmonary artery stenosis in Alagille syndrome. Most (but not all) centres perform a percutaneous liver biopsy as a final step in the diagnostic analysis. In case of sufficient histological indications compatible with biliary obstruction, particularly if combined with ductular reaction and fibrosis, the decision is frequently made to perform intraoperative cholangiography, followed by KPE, based on positive cholangiography or macroscopic absence of extrahepatic bile ducts. The diagnostic use of hepatobiliary scintigraphy for the discrimination of biliary atresia is no longer advocated, partly because of relatively low specificity [19] which fails to abolish the need for liver biopsy and, if suspicious, an intraoperative cholangiogram.

### 3.2. Surgical Strategy (Based on 42 Completed Questionnaires)

Confirmation of the actual diagnosis is the first step in the operation, and 80% of respondents indicated that they would do this using a small right upper quadrant incision centred over the anticipated position of the gallbladder; 20% would utilize less invasive techniques for this step. Cholangiography was regarded as “obligatory” at this stage by 60% of practitioners surveyed, by 33% only when inspection of the hepatoduodenal ligament was inconclusive, and by 7% when an ERCP was not available or was inconclusive. The use of indocyanine green is becoming more prevalent in adult biliary surgery, and one respondent stated that they would use this diagnostically during KPE. Following confirmation of the diagnosis, 95% would then look for other features of syndromic BA (e.g., polysplenia).

There was only a single respondent who would perform the KPE laparoscopically, all others opting for a conventional open technique. With this in mind, the degree of liver mobilisation prior to porta hepatis was then investigated. Total mobilisation (i.e., division of suspensory ligaments on both right and left) would be performed by 33%, partial mobilisation (i.e., division of only falciform and left triangular ligaments) by 43%, and no additional mobilisation by 24%.

Surgical loupe magnification was used by 90% of respondents during dissection of the porta hepatis, with most carrying out sharp dissection using a knife (78%) and avoiding bipolar diathermy (52%).

Deliberate exposure of the Rex recessus was sought by 52%, and was regarded as optional by 14%. Most surgeons (67%) aimed for an extended dissection of the porta hepatis (beyond the division of the portal vein), with the remainder actively avoiding this.

The preferred length of the retrocolic Roux loop was 25–50 cm for 90% of respondents, but 5% aimed for less than this while 5% aimed for more. A single respondent stated that they would create an intussusception valve within the loop. Most surgeons would use a hand-sewn (86%), end-to-side technique (67%) for the jejunojejunostomy.

All surgeons performed an end-to-side portoenterostomy using either an interrupted (48%) or running stitch (43%) or both (9%), and usually using 6/0 sutures (57%). The total time for the KPE was assessed at 120–240 min (76%) and for those performing it laparoscopically >240 min (71%).

Post-operatively, most respondents felt that prophylactic antibiotics should be prescribed for >1 week (82%) and the regimen should include ursodeoxycholic acid (94%). Respondents were split between use of steroids (50%) and no one used immunoglobulins, farnesoid X receptor agonists, or ileal bile acid transporter (IBAT) inhibitors.

## 4. Discussion

The main aim of this paper is to present the breadth of current surgical techniques and practice for biliary atresia as performed in a predominantly European and North American setting. We did not seek to debate actual indications or limitations such as the late-presenting infant or the possible role of liver transplantation as a primary procedure.

### 4.1. Surgical Strategy

Our survey confirms an important observation which bears repeating, that the Kasai portoenterostomy is not a single, uniform operation performed every time by each surgeon in the same way. Rather, it is simply a principle of extrahepatic dissection and excision with (nowadays) a Roux loop reconstruction and anastomosis high in the porta hepatis. It might be questioned whether the details matter, and in some respects they probably do not. Visualisation of the biliary tree is an axiom of diagnosis, but can be achieved as easily by laparoscopy as by an exploratory right upper quadrant incision. Similarly, despite textbook adherence to the concept of cholangiography, in most cases the gallbladder is so atrophic as to be without a lumen and hence the siting of a catheter is not possible. Alternatively, about 20% of cases have a gallbladder that is structurally normal (evident on ultrasound) but is filled with clear mucus. Cholangiography is essentially redundant here, in that it will inevitably show a patent common bile duct into the duodenum but no sign of a more proximal biliary tree. Some centres, particularly in France, would consider a portocholecystostomy (i.e., the gallbladder opened and anastomosed to the transected portal plate) for BA to make use of this feature [20]. This option effectively abolishes the risk of post-operative cholangitis, although has a higher revision rate for leaks and obstruction.

In those centres with a less stringent, less discriminatory pre-surgical workup, the proportion of non-BA cases coming to surgical exploration is likely to be higher and cholangiography to exclude BA will performed more frequently. Cholangiography is also important in cases of cystic BA, where it may or may not show connections to the residual intrahepatic ducts and is able to distinguish cystic BA from cystic choledochal malformation—a much more benign condition.

#### 4.1.1. Role of Laparoscopy

Laparoscopic KPE remains unpopular in Europe and North America, although some of the early reports arose from European centres [21,22,23]. More recently, more reports and interest have arisen in the larger Asian centres, including in China [11,24]. However, no report has shown or even implied its superiority to the more conventional open approach, and indeed it is difficult to see a real rationale for this technique beyond the cosmetic. Certainly, in terms of the primary objective—clearance of jaundice and preservation of the native liver—it clearly has no advantage.

#### 4.1.2. Degree of Liver Mobilisation

The need for liver mobilisation in the open technique continues to be contentious, but exteriorization was favoured by 75% of respondents, although it is clear that this can be achieved by less than complete mobilisation, dividing only the falciform ± left triangular ligaments—the most popular technique. The degree of dissection in the porta hepatis has changed over the years. It is clear from Kasai’s original descriptions that he was relatively conservative in leaving residual biliary tissue in the porta and performing the Roux loop anastomosis to form an ovoid section. His successor in Sendai, Ryoji Ohi, and other Japanese surgeons [25] adopted more radical dissection to widen the resultant portoenterostomy and incorporate the interstices behind the bifurcation of the portal vein on the left into the Rex recessus (facilitated by dividing the bridge between the third and fourth segments), and on the right up to and sometimes into Rouviere’s fosse (containing the posterior branches of the right hepatic artery and right portal vein). The majority (67%) of our surgical group seem to favour this latter approach. Interestingly, Ohi’s successor in Sendai, Masaki Nio, reverted somewhat to a less extensive dissection more in keeping with the original [26]. Nowadays the only group of surgeons who have reverted to Kasai’s original approach have been those who do this operation entirely laparoscopically, as it is evident that radical dissection is only effectively possible as an open technique [27].

#### 4.1.3. Roux Loop

Use of a retrocolic Roux loop was standard for all respondents, with the only debate being about how long it should be. The clear majority aim for 40 cm with only one respondent aiming for a short (<25 cm) loop and one deliberately measuring it as >50 cm. There is little firm evidence on this matter presented in the literature. Most recently, a Chinese centre prospectively randomized 166 infants, comparing standard length with a short length (13–20 cm) [28]; there was no difference in incidence of cholangitis (43% vs. 47%) or clearance of jaundice (45% vs. 50%).

A single surgeon favoured modification of the Roux limb by creation of an “intussusception valve”. These were briefly in vogue during the 1990s [9] as they were thought to prevent reflux and hence reduce cholangitis. A small-scale prospective trial from Tokyo involving 20 infants showed no difference in outcome [29] and interest seemed to wane. However, more recently this technique has become popular, at least in China where a recent questionnaire-based survey showed it was being used in half of their centres [30]. It is unclear whether this is in any way evidence-based.

A range of different techniques were reported for the jejunojejunostomy, including end-to-side, end-to-end, hand-sewn, and stapled. By contrast, the actual portoenterostomy anastomosis was invariably end-to-side and fashioned by most contributors using a relatively fine (6/0) absorbable suture (PDS).

### 4.2. Post Operative Adjuvant Therapy

Biliary atresia affects both intra- and extra-hepatic bile ducts, and KPE primarily treats only the extra-hepatic component. Many medications have been advocated to treat the intra-hepatic bile duct damage, though few with any substantial evidence base.

The liver in the BA infant prior to KPE is subject to substantial cholestatic injury with subsequent fibroinflammatory and necrotic pathophysiological adaptations [31]. KPE provides surgical relief of the extrahepatic obstruction, which is key for the BA liver to begin the process attempt to restore normal homeostasis, including reduction of retained bile acids [32], and subsequent activity related to the presence of activated profibrotic and inflammatory mechanisms [33,34,35]. However, despite KPE, the majority of infants with BA present ongoing cholestasis and progression of liver disease, underlying the expectation of continued inherent developmental cholangiopathy in these patients. In brief, the expected early results of KPE are native liver survival (NLS) at age 2 of ~45–65%, with significant variability between centres and countries e.g., [36,37,38,39,40,41]. Moreover, progressive liver disease is ongoing in BA during childhood, leading ultimately to NLS rates of ~25% by the beginnings of adulthood [39,41,42,43]. Thus, with progression of disease and the rapid progression of subsequent liver disease there is a need for medical therapies aimed at improving outcomes after KPE, in order to reduce the risk of death and the need for liver transplant [44]. Among the adjuvant therapies that are generally accepted internationally are antibiotics to address the risk of cholangitis, and ursodeoxycholic acid (UDCA) to act as a potential choleretic and to improve the hydrophilicity of the bile acid pool [44,45]. UDCA and antibiotics both received strong support in our survey (>80% of 33 respondents).

The use of corticosteroids post-KPE is among the more controversial aspects of treatment. Large studies have supported its use [46], including a recent randomised trial from Shanghai [47]. However, there have also been studies indicating no discernible benefit according to the available evidence [48]. Despite this gap in therapeutic support for steroids post-KPE, along with evidence of an increase in side effects and impaired growth among patients receiving steroids [49]; half (17/34) of respondents to the survey reported their use. This is a crucial issue, as use of corticosteroids post-KPE varies widely around the globe—from 0–100%. Clearly, more investigations are needed.

There are currently two ongoing international investigational studies exploring the efficacy of intestinal bile acid transporter (IBAT) inhibitors post-KPE. These studies are focussed on maralixibat (Phase 2, NCT 04524390; EMBARK), assessing a short-term outcome of reduction in total bilirubin at 6 months, and odevixibat (Phase 3 NCT 04336722; BOLD), with a clinical outcome measure of improvement of NLS at 2 years. The rationale for IBAT inhibition seeks to address one of the causes of liver damage in cholestatic diseases such as BA, by reducing the obligate intrahepatic bile acid levels [50]. IBAT inhibitors are approved in many countries for relief of pruritus in children with Alagille syndrome and PFIC [50,51,52].

Intriguingly, there was support for determining coexistent CMV infection at the time of KPE, which should lead to consideration for antiviral treatment [53], with the potential to improve post-KPE outcomes. Other considerations for improving outcomes that have yet to be studied include varying methods of feeding infants (e.g., breast milk versus formula), the role of parenteral nutrition, and appropriate roles for fat-soluble vitamin monitoring and supplementation.

At least three-quarters of BA patients will need LT in their lifetime [39,54,55]. KPE is essentially palliative in nature and is typically only the first step of treatment. One element that was not addressed in our survey was the prevention of complications evident at the time of transplantation (i.e., reducing post-operative intra-abdominal adhesion). It has been shown [56,57] that duration of total hepatectomy, bleeding, and prognosis of liver transplantation were adversely impacted when KPE was performed in a centre that does not perform liver transplantation. Operative details which may reduce adhesions include avoidance of unnecessary exposure of the intestines outside the abdomen, avoidance of abdominal drains, and possibly the use of anti-adhesion adjuncts such as hyaluronic acid (Seprafilm™, Baxter Inc., Deerfield, IL, USA) and hydrogels (CoSeal™, Baxter Inc., Deerfeield, IL, USA) around the liver. Both are in current use in European centres, though without evidential support.

The obvious limitation of this study is that it is divorced from any report of actual outcomes, and remains opinion-based. Nevertheless, it reflects the views of a large body of surgeons and clinicians involved in the care of these infants.

## 5. Conclusions and Future Directions

In conclusion, the lack of uniform approach and absence of registries hampers progress in untangling the nuances of the Kasai operation and determining the role of adjuvant therapies for this complex and perplexing disease. It also seems very unlikely that sufficiently powered randomized trials will be available to arrive at a clear answer. Nevertheless, we look forward to a time in the near future when testing of existing and novel therapies will be readily available to help guide clinicians and parents to achieve optimal outcomes. Furthermore, we note the emergence of possible biomarkers which may better refine the diagnostic (e.g., MMP-7) [58] or prognostic processes (e.g., secretin receptor expression) [59].

## Data Availability

Not applicable.

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
