# Peer review of "Surgical and Medical Aspects of the Initial Treatment of Biliary Atresia: Position Paper"

_jcm, 2022, doi:10.3390/jcm11216601_

Round 1
Reviewer 1 Report (Previous Reviewer 2)
Although the Manuscript is informative and has been much improved through revision, I still think the author need add some contents about diagnostic or screening toos for BA, such as serum MMP-7, serum direct bilirubin within three days of birth and AI model based on sonographic gallbladder images(as I mentioned previously), in INTRODUCTION. Because the potential readers of the JCM are not only pediatric surgeons but also those who work in general medicine.
Author Response
Response to Reviewer #1
Thank-you for reconsideration of our article. I have added two further paragraphs in the Introduction on the points you raised with four further references.
- Jiang J, Wang J, Shen Z et al. Serum MMP-7 in the diagnosis of biliary atresia. Pediatrics 2019; 144: e20190902
- Wu JF, Jeng YM, Chen HL, et al. Quantification of serum matrix metallopeptide 7 levels may assist in the diagnosis and predict the outcome for patients with biliary atresia. J Pediatr. 2019;208:30–37.e1
- Hsu FR, Dai ST, Chou CM, Huang SY. The application of artificial intelligence to support biliary atresia screening by ultrasound images: A study based on deep learning models. PLoS One. 2022; 19;17:e0276278. doi: 10.1371/journal.pone.0276278.
- Harpavat S, Garcia-Prats JA, Anaya C, Brandt ML et al. Diagnostic yield of newborn screening for biliary atresia using direct or conjugated bilirubin measurements. JAMA. 2020;323:1141-1150. doi: 10.1001/jama.2020.0837.
Reviewer 2 Report (New Reviewer)
I appreciate that this summary of the questionnaire carried out at BARD 2021 is shared nicely with the rest of the world. Just one point, however, the historical aspect in the degree of dissection in the porta hepatis among Japanese surgeons that the authors described (lines 203-215) is correct but that is not all. Therefore, I would like to share the story remaining, which I think is important among surgeons in the rest of the world.
The next successor in Sendai after Ryoji Ohi was Masaki Nio, who made the portal dissection reverted, in other words, less radical, backward to Kasai’s original level because the outcome of Ohi’s extensive dissection was retrospectively not so good as original Kasai’s series in Sendai. This experience has been shared among Japanese surgeons and today, most of the Japanese pediatric surgeons, actually even some of the laparoscopic surgeons understand and practice the nuance of this level, which is on the left into Recessus of Rex and on the right up to or probably more often just before Rouviere’s fossa. This was described in the literature; Nio M, Wada M, Sasaki H, et al.: Technical Standardization of Kasai Portoenterostomy in Biliary Atresia. J Pediatr Surg. 51:2105- 2108, 2016.
Author Response
Response to Reviewer #2
Thank you for your review.
It was really difficult to actually explore the nuances of individual “Kasai-like” surgery such as you describe given it was based on questionnaire and with no way of validating actual technique itself. Nevertheless I am happy to add this further point in the evolution of the operation, though the paper actually compares four groups, with the early ones have double Roux loops, Suruga modifications and valves etc.
I have added the reference suggested.
Nio M, Wada M, Sasaki H, et al.: Technical Standardization of Kasai Portoenterostomy in Biliary Atresia. J Pediatr Surg. 51:2105- 2108, 2016.
Reviewer 3 Report (New Reviewer)
This seems to be a result of the questionnaires performed concerning biliary atresia. There are major problems concerning this manuscript.
1. The abstract is too short and I cannot understand the methods or results of this manuscript.
2. I don't think the style of discussion section is appropriate.
3. Please explain whether or not this is a review paper or an original study.
Author Response
Response to Reviewer #3
We thank you for your review. Please note that this paper has been through two rounds and previous reviewers have not raised any of the comments on structure that you highlight.
This seems to be a result of the questionnaires performed concerning biliary atresia. There are major problems concerning this manuscript.
- The abstract is too short and I cannot understand the methods or results of this manuscript.
I take this point and I have increased the length of the abstract to try and explain the methods used (questionnaire; expert-panel commentary). I did not think it appropriate, however, to list the results to each and every question here.
Full details of the Methods used and the actual questionnaire are given in the text and a supplemental file. Similarly the results are listed in the appropriate section. Section 3.1 was expert-panel based. Section 3.2 was questionnaire-based.
- I don't think the style of discussion section is appropriate.
We sectioned this (e.g. Surgical Strategy, Role of Laparoscopy etc.) to allow meaningful discussion of all the points considered in the Results section in a logical order.
- Please explain whether or not this is a review paper or an original study.
This has elements of both. The questionnaire is clearly original and was analysed as such with the results (6.2) being reviewed and discussed during the lead-up to the virtual BARD webinar of 2021. The Discussion provides review-quality context of medical, diagnostic and surgical issues raised by the questionnaire. Previous reviewers also felt the need to expand outside of the areas which we have complied with (e.g. role of screening). This article is one of a series of publications in the Journal of Clinical Medicine based on the 2021 and then 2022 expert-led BARD Congress which do use this expert panel-based approach with literature review. Please see:
- Calinescu AM, Madadi-Sanjani O, Mack C, Schreiber RA, Superina R, Kelly D, Petersen C, Wildhaber BE. Cholangitis Definition and Treatment after Kasai Hepatoportoenterostomy for Biliary Atresia: A Delphi Process and International Expert Panel. J Clin Med. 2022 Jan 19;11(3):494. doi: 10.3390/jcm11030494. PMID: 35159946; PMCID: PMC8836553.
- Fischler B, Czubkowski P, Dezsofi A, Liliemark U, Socha P, Sokol RJ, Svensson JF, Davenport M. Incidence, Impact and Treatment of Ongoing CMV Infection in Patients with Biliary Atresia in Four European Centres. J Clin Med. 2022 Feb 11;11(4):945. doi: 10.3390/jcm11040945. PMID: 35207217; PMCID: PMC8879500.
- de Ville de Goyet J, Illhardt T, Chardot C, Dike PN, Baumann U, Brandt K, Wildhaber BE, Pakarinen M, di Francesco F, Sturm E, Cornet M, Lemoine C, Pfister ED, Calinescu AM, Hukkinen M, Harpavat S, Tuzzolino F, Superina R. Variability of Care and Access to Transplantation for Children with Biliary Atresia Who Need a Liver Replacement. J Clin Med. 2022 Apr 12;11(8):2142. doi: 10.3390/jcm11082142. PMID: 35456234; PMCID: PMC9032543.
Round 2
Reviewer 3 Report (New Reviewer)
I think reply letter would be necessary.
This manuscript is a resubmission of an earlier submission. The following is a list of the peer review reports and author responses from that submission.
Round 1
Reviewer 1 Report
Authors: not clear if the authors should include all the working group? Cincinnati is mis-spelled (Cincinnatti)
Structure
1. Introduction. 'accompanying' cholangiography-- the preferred term in 'intraoperative'. The last paragraph on liver transplantation is beyond the scope of this introduction and is not relevant - suggest removing. Perhaps, given the authors' seniority, they might want to discuss future areas of research as one of the aims?
2. Material and methods: suggest survey questions be outlined or at the very least topics covered. 'The 'oucomes' of the questionnaire...'-- correct wording should be 'results'. '...were subsequently discussed....'-- rather: 'analyzed??''
3. Results:
Diagnostic strategy- this section needs figures or graphs of the results of the survey. The discussion about the role of genetic analyses and their turn-around time does not belong in the results section unless this was a question in the survey- either way, the discussion does not belong here.
The paragraph beginning with 'Generally, the diagnostic strategy....'--it is not clear if this paragraph is a result or merely a summary of what is recommended in ref 12- to which the reader is referred.
Surgical Strategy: this section is better and more structured although the data would benefit from a graphical representation
4. Discussion: the discussion does not cover all the aims, notably not the diagnostic strategy. In addition, results are included here (the Roux loop paragraph) rather than in the results section. The discussion would benefit from sub-headings (mobilization, Roux loop, jejunojejunostomy...)
The paragraph 'Early performance....' -- not clear to this reviewer why it is here. Did one of the questions address timing of KPE?
Line 286-289: was nutrition a question? if so, should be reported in results rather than new concept in discussion
Conclusion: no mention of optimal surgical approach, only focused on adjuvant therapies.
Syntax: several extra spaces or . in the wrong place.
Author Response
RESPONSE TO REVIEWS
REVIEWER #1
Authors: not clear if the authors should include all the working group?
Not as authors, they had no input to the manuscript.
Cincinnati is mis-spelled (Cincinnatti)
Is corrected in the text
Structure
- Introduction. 'accompanying' cholangiography-- the preferred term in 'intraoperative'. The last paragraph on liver transplantation is beyond the scope of this introduction and is not relevant - suggest removing. Perhaps, given the authors' seniority, they might want to discuss future areas of research as one of the aims?
We feel that the “Transplantation” paragraph completes the “story” of BA and would look odd to the general reader without it. I have added a sentence of research aims.
- Material and methods: suggest survey questions be outlined or at the very least topics covered. 'The 'oucomes' of the questionnaire...'-- correct wording should be 'results'. '...were subsequently discussed....'-- rather: 'analyzed??''
Changed for “results” but they were discussed actually – as there was not much to analyse. For reference I have attached a summary of the questions as a supplemental file.
- Results:
Diagnostic strategy- this section needs figures or graphs of the results of the survey. The discussion about the role of genetic analyses and their turn-around time does not belong in the results section unless this was a question in the survey- either way, the discussion does not belong here.
The questionnaire actually had very little about the diagnostic process and these “results” (section 3.1) are the result of the discussions but principally the workshop that followed. We did feel it important to reflect this here as a consensus outcome. We have no “figures/graphs” to display for this reason.
The paragraph beginning with 'Generally, the diagnostic strategy....'--it is not clear if this paragraph is a result or merely a summary of what is recommended in ref 12- to which the reader is referred.
The relevant hepatology authors [SK, HV] on our paper contributed to formation of those guidelines [12] and felt strongly that they did not want to go beyond it or indeed start from scratch. So, yes, it is more of a summary.
Surgical Strategy: this section is better and more structured although the data would benefit from a graphical representation
Any graphs we generated for the workshop were very simple bar graphs conveying no more details than is in the text realistically.
- Discussion: the discussion does not cover all the aims, notably not the diagnostic strategy. In addition, results are included here (the Roux loop paragraph) rather than in the results section. The discussion would benefit from sub-headings (mobilization, Roux loop, jejunojejunostomy...)
This is true and it reflects that, as you have already highlighted, 3.1 is both result and discussion. I have left as is, but equally if directed could put the entire section into the discussion.
I have introduced section sub-headings in the discussion.
The paragraph 'Early performance....' -- not clear to this reviewer why it is here. Did one of the questions address timing of KPE?
It’s a dogmatic statement, which I agree was not really part of the questionnaire.. I have removed it.
Line 286-289: was nutrition a question? if so, should be reported in results rather than new concept in discussion.
It was not a question but again came up in the discussion and the medical members felt it important to continue to re-iterate.
Conclusion: no mention of optimal surgical approach, only focused on adjuvant therapies.
Well I am not quite sure what we all mean by an “optimal surgical approach”; the results really reflect that there is a range of possible approaches being practised and they all think it is optimal.
Syntax: several extra spaces or . in the wrong place.
Reviewer 2 Report
The manuscript presents the results of a questionnaire about BA managements from international experts. It is no doubt valuable to publish on JCM. Nevertheless, I think it should be published as a review article rather than an original article, because it is all about opinions but scientific conclusions.
Here is my comment:
In the past several years, several new diagnostic or screening tools for BA have been investigated and showed good performance, such as serum MMP-7, serum direct bilirubin within three days of birth and AI model based on sonographic gallbladder images. It is better to mention these in the Introduction and add some comments in the Discussion.
Here are papers about these tools:
For MMP-7
Lertudomphonwanit, C. et al. Large-scale proteomics identifies MMP-7 as a sentinel of epithelial injury and of biliary atresia. Sci. Transl. Med. 9, eaan8462 (2017);
Yang, L. et al. Diagnostic accuracy of serum matrix metalloproteinase-7 for biliary atresia. Hepatology 68, 2069–2077 (2018)
For bilirubin screening:
Harpavat, S., Garcia-Prats, J. A. & Shneider, B. L. Newborn bilirubin screening for biliary atresia. N. Engl. J. Med. 375, 605–606 (2016);
Harpavat, S. et al. Diagnostic yield of newborn screening for biliary atresia using direct or conjugated bilirubin measurements. JAMA 323, 1141–1150 (2020)
For AI model based on sonographic gallbladder images:
Zhou, W., Yang, Y., Yu, C. et al. Ensembled deep learning model outperforms human experts in diagnosing biliary atresia from sonographic gallbladder images. Nat Commun 12, 1259 (2021). https://doi.org/10.1038/s41467-021-21466-z
Author Response
RESPONSE TO REVEWS
REVIEWER #2
The manuscript presents the results of a questionnaire about BA managements from international experts. It is no doubt valuable to publish on JCM. Nevertheless, I think it should be published as a review article rather than an original article, because it is all about opinions but scientific conclusions.
To some extent we agree with this sentiment and have written it as such and it is in the limitations section of the discussion. It is all about opinions but we have at least a structured questionnaire as the basis for further discussion.
Here is my comment:
In the past several years, several new diagnostic or screening tools for BA have been investigated and showed good performance, such as serum MMP-7, serum direct bilirubin within three days of birth and AI model based on sonographic gallbladder images. It is better to mention these in the Introduction and add some comments in the Discussion.
We limited discussion on role or value of Screening (not in questionnaire- d1-3 direct bilirubin only happens in Texas- though its principle advocate was part of the webinar and it was part of a separate presentation in the Webinar and I believe will also be a review in JCM); MMP7 (contentious and not done routinely in Europe or North America, although again its principle advocate was part of the webinar). AI model of gallbladder images remains a long way from any centre involved in these discussions – it’s a Chinese thing. I have added a sentence at the end on future directions which includes some of these suggestions though.
